# Mosquitocidal Chips Containing the Insect Growth Regulator Pyriproxyfen for Control of *Aedes aegypti* (Diptera: Culicidae)

**DOI:** 10.3390/ijerph16122152

**Published:** 2019-06-18

**Authors:** Kristen C. Stevens, Roberto M. Pereira, Philip G. Koehler

**Affiliations:** Entomology and Nematology Department, University of Florida, Gainesville, FL 32653, USA; kcstevens93@gmail.com (K.C.S.); pgk@ufl.edu (P.G.K.)

**Keywords:** Mosquito, *Aedes aegypti*, Diptera, Culicidae, Insect Growth Regulator, Pyriproxyfen

## Abstract

*Aedes aegypti* were exposed to water treated with mosquitocidal chips containing the insecticide pyriproxyfen in a polymer formulation. Chips were tested under different conditions; different water volumes, in containers made of different material, and in water with different levels of organic matter. Treated chips caused 100% mortality of *Ae. aegypti* during their pupal stage independent of size or type of container, and the mount of organic matter contained in the water to which the mosquito larvae were exposed. When mosquitocidal chips were used in >25% of the oviposition containers within their cages, there was a significant control of the mosquito populations. Mosquitocidal chips worked in different environments, caused significant mosquito population decreases, and were effective in controlling *Ae. aegypti.*

## 1. Introduction

*Aedes aegypti*, the yellow fever mosquito, is considered one of the world’s greatest health threats. One class of chemicals commonly used to control *Ae. aegypti* larvae are insect growth regulators (IGRs) [1,2]. IGRs disrupt insect growth and reproduction by interfering with insect development [1,2,3]. Juvenile hormone analogs (JHAs) disrupt insect development and prevent insects from reaching the adult stage by providing increasing juvenile hormone in insects at a time these compounds do not normally occur [2,3], therefore preventing proper mosquito development.

Pyriproxyfen, a juvenile hormone analog, is a relatively stable chemical which results in insects being unable to molt to the adult stage [3,4]. It is approved by the U.S. Environmental Protection Agency for use in small containers to control *Ae. aegypti* because of its relatively low toxicity to non-target organisms [4,5,6], and safety to humans. The World Health Organization (WHO) has also approved pyriproxyfen at a rate of 10 PPB for use in potable water [7].

Control of *Ae. aegypti* is difficult because of different behaviors including daytime feeding habits, the ability to develop in a wide variety of water holding containers, and skip oviposition, where one female will lay her eggs in numerous containers [8,9,10]. However, pyriproxyfen is effective at reducing populations of *Ae. aegypti* [11,12] since *Ae. aegypti* females are not deterred from laying eggs in pyriproxyfen-treated containers.

Although effective for mosquito control, pyriproxyfen is labeled for treating large bodies of water, complicating control in small containers. The objective of this study was to test the efficacy of mosquitocidal chips treated with slow-release pyriproxyfen formulation to decrease *Ae. aegypti* populations. Use of pre-treated chips with a small dose of pyriproxyfen facilitates the use of the product by final users, especially those lacking formal education. The pre-dosed chip requires no preparation, dilution, or additional manipulation by the final user, who can apply the mosquitocidal chips to a variety of potential mosquito breeding containers. Although the use of pyriproxyfen in drinking water may not be advisable due to some health concerns [13], its use in containers of water not destined to human consumption, or other natural or artificial water-holding vessels, can be an important tool in the control of mosquitoes around human dwellings. The slow-release formulation on the chips remains active for a minimum of 6 months through dry–wet cycles and can provide continuous control of mosquitoes for a whole mosquito for at least one season without reapplication of the product.

## 2. Materials and Methods

*Aedes aegypti* colony with no known resistance to insecticides was acquired from the Center of Medical, Agricultural and Veterinary Entomology (CMAVE) and the United States Department of Agriculture, Agricultural Research Service (USDA-ARS), Gainesville, FL. The colony was maintained in 30 cm × 30 cm × 30 cm cages (BioQuip^®^ Lumite Screen Collapsible Cages, Rancho Dominguez, CA, USA), provided with 10% sugar solution and tap water in a rearing room maintained at 28 ± 2 °C with a relative humidity of 36% ± 5% and a 12:12 (L:D) photoperiod. Female mosquitoes were blood-fed on live domestic chickens (IACUC Protocol #20163836_01).

Mosquito eggs were hatched by placing strips of dried egg sheets into 55 cm × 45 cm × 8 cm into plastic trays (Del-Tec/Panel Control Plastic Trays, Greenville, SC, USA) with clean unchlorinated water. Larvae were fed ground fish flakes (TetraFin^®^ Goldfish Flakes, Blacksburg, VA, USA). Pupae were placed into polypropylene (450 mL) cups and put into adult rearing cages for emergence.

Technical grade pyriproxyfen (Nylar^®^ Technical, MGK^®^ Insect Control Solutions, Minneapolis, MN, USA) was dissolved in methanol and serial diluted as needed for experiments or formulated for application on mosquitocidal chips. A chip formulation, containing 0.01% pyriproxyfen, was prepared using a base formulation with 1% fumed silica, 5% Butyl-methacrylate polymer, and 94% acetone. The application of 100 µL of the 0.01% pyriproxyfen formulation delivered 8.4 μg of the active ingredient to the chip. The control formulation contained all the formulations ingredients but no pyriproxyfen.

Ceramic tiles (hexagonal with 8mm side) (American Olean Satinglo Hex Honeycomb Mosaic Ceramic Floor and Wall Tile, Birmingham, AL, USA) were removed from glue backing and cleaned with dish soap and warm water and dried before being treated with the mosquitocidal formulations using a micropipette. Mosquitocidal chips were treated using 100 μL of the stock pyriproxyfen formulations pipetted onto the chips. Control chips received formulation with no active ingredient. Formulations were applied to the non-glazed side of each chip to ensure treatments adhered to the tile. Mosquitocidal chips were allowed to dry for a minimum of 24 h in a chemical hood before being placed in bioassay containers.

Polypropylene cups (WNA^TM^, Chattanooga, TN, USA, 450 mL) were filled with 350 mL of clean unchlorinated water and treated with pyriproxyfen-treated or control chips. Ten late 3rd to 4th instar *Ae. aegypti* larvae were added to each bioassay container.

The purpose of the water volume experiment was to determine whether mosquitocidal chips effects were affected by varying water volumes (250, 500, 750 and 1000 mL) of clean unchlorinated water. Treatment and control vases (1000 mL, Libbey^®^ glass cylinder vase, Toledo, OH, USA) contained the same volumes of water and untreated chips, or 8.4 μg pyriproxyfen chips (0.01% pyriproxyfen), which were deposited on the bottom of the vases using large forceps. Ten late 3rd–early 4th instar mosquitoes were pipetted into each vase from their rearing cups. There were four replicates of each treatment and control. Larvae were fed 200 μL of a slurry of ground fish food every other day. Vases were maintained at ca. 31 °C and 15% relative humidity (RH) and inspected every 24 h for dead or live larvae, pupae and adults. Experiments were run for 4 d or until all mosquitoes had either died or emerged as adults.

Percent mortality of dead insects in experiments was calculated and then arcsin-transformed, an analyzed using repeated measures ANOVA using days after application as the repeated measure. Mean mortalities were compared using a Tukey’s Honest Significant Difference (HSD) pairwise comparison.

The purpose of the container material experiment was to determine whether mosquitocidal chips were affected by container material that simulated habitats where *Ae. aegypti* larvae typically develop. The materials used were wood (Artminds^®^ wooden box, Southfield, MI, USA), metal (Ashland^®^ Galvanized Metal Bucket, Ashland, Covington, KY, USA), clay (Indigo spice, studio décor, Irving, TX, USA), ceramic (Indigo spice, studio décor, Irving, TX, USA), plastic 450 mL polypropylene cups (WNA^TM^, Chattanooga, TN, USA), and glass (Kimble^®^ Wide Mouth Jars, Vineland, NJ, USA). Unchlorinated water (200 mL) was placed into each container with either an 8.4 μg pyriproxyfen treated (0.01% pyriproxyfen) or an untreated chip. Wood containers were tightly wrapped with a layer of parafilm in order to prevent leakage for the duration of the experiment. Ten late 3rd–early 4th instar mosquitoes were placed in each of four replicates of each container type. There were two replicates of each container type with control chips. Insects were maintained and checked as described above. Percent mortality of insects was calculated and analyzed, as described above.

The organic matter experiment was designed to determine if different percentages of organic matter in water would affect chip efficacy. Treatments included 350 mL of water containing either 0%, 10%, 30%, 50%, 70% and 90% of a leaf infusion prepared according to Reiter et al. [14] and the 8.4 μg pyriproxyfen chip (0.01% pyriproxyfen), or control chip for control treatments. Ten late 3rd–early 4th instar mosquitoes were placed in each cup and four replicates were prepared for each treatment and control. Insects were maintained and checked as described above. Percent mortality of insects was calculated and analyzed as described above.

The effects of the presence of mosquitocidal chips on female oviposition preference and on the overall reduction of populations of *Ae. aegypti* were tested using cages (60 cm × 60 cm × 60 cm BugDorm Insect Tents, MegaView Science Co., Ltd., Taichung, Taiwan, China), containing 4 cups with 350 mL of clean unchlorinated water with an oak leaf sachet prepared with fillable tea bags (disposable, self-seal tea bags, Otter and Trout Trading Co, Gainesville, FL, USA), containing 0.5 g of ground field-collected oak leaves. Cups were lined internally with filter paper where female mosquitoes could oviposit eggs. There were 4 treatments were: (a) 3 untreated cups and 1 treated cup with an 8.4 µg pyriproxyfen chip (0.01% pyriproxyfen), (b) 2 untreated cups and 2 cups treated with pyriproxyfen chips, (c) 1 untreated cup and 3 cups treated with pyriproxyfen chips, and (d) 4 cups treated with pyriproxyfen chips. For the controls, all 4 cups were untreated. There were 4 replicates of each of the 4 treatments and control, and the experiment was repeated twice over a 2 month period.

Ten gravid female *Ae. aegypti* were put into each cage 48 h after blood feeding and were allowed to oviposit on filter paper for 72 h. After this time, egg sheets and adult mosquitoes were removed from the cage. Egg sheets were allowed to dry for 24 h and eggs were counted, removed from the papers and returned to their original containers. Chips were temporarily removed from the containers which were closed with lids and hand shaken for 1 min to stimulate egg hatching. After shaking, the lid was removed, and chips were placed back into original containers. Larvae in containers were fed 200 µL of ground fish food every other day. A 120 mL cup with 10% sugar solution was placed in each cage for emerging adult mosquitoes to feed on. After 10 d, emerged adults were counted. Experiments were kept in a greenhouse at ca. 35 °C ± 5 °C and 25% ± 5 °C RH with a photoperiod between 12:12 (L:D) and 14:12 (L:D). Percent emergence data was calculated by using the number of eggs laid and number of adults emerged and was arcsin-transformed for statistical analysis. Number of eggs laid and number of adults emerged were analyzed using a one-way ANOVA and percent emergence data were compared using a Student’s t-test.

## 3. Results

### 3.1. Water Volume

There was a significant difference in times to mortality (F = 261.2, df = 3, *p* ≤ 0.0001, Figure 1) and in mosquito mortality (F = 96.74, df = 4, *p* ≤ 0.0001) when mosquitocidal chips were used in different water volumes and a significant interaction between water volume and time (F = 17.05, df = 12, *p* ≤ 0.0001). Pairwise comparisons among water volumes showed that mosquito larvae exposed to chips in 250 mL of water died at significantly faster rates than mosquito larvae exposed to the chips in all other water volumes (500 mL: *p* ≤ 0.0028, 750 mL *p* = 0.0002, 1000 mL *p* = 0.0174). However, 100% mortality was observed on fourth day for all treatments. The larvae in 250 mL treatment reached 100% mortality 24 h prior to larvae exposed to at all other volumes.

### 3.2. Container Material

There was a significant effect of container materials (F = 16.95, df = 5, *p* ≤ 0.0001), time (F = 609.35, df = 3, *p* ≤ 0.0001) and material–time interaction (F = 7.12, df = 15, *p* ≤ 0.0001) (Figure 2). No significant difference in mosquito mortality was observed between ceramic and clay containers (t = 1.35, df = 35, *p* = 0.755); however, mosquitoes in ceramic containers died at a significantly faster rate than glass (t = 4.51, df= 35, *p* = 0.0009); metal (t= 7.96, df = 35, *p* ≤ 0.0001); plastic (t = 5.59, df = 35, *p* ≤ 0.0001); and wood containers (t = 4.95, df = 35, *p* = 0.0003). Additionally, clay containers had a significantly faster mosquito mortality rate than glass (t = 3.16, df = 35, *p* = 0.035); metal (t = 6.61, df = 35, *p* ≤ 0.0001); and plastic (t = 4.24, df = 35, *p* = 0.002); and wood (t = 3.60, df = 35, *p* = 0.012). Mosquitoes in glass containers had significantly faster mortality than mosquitoes in metal containers (t = 3.45, df = 35, *p* = 0.0170,). No other treatments comparisons showed any significant difference in mosquito mortality rate.

### 3.3. Effects of Organic Matter

There was no significant difference in mosquito mortality with different percentages of leaf infusion (F = 0.422, df = 5, *p* ≤ 0.829, Figure 3), although in the 0% and 10% leaf infusions, mosquitoes were killed at a faster rate than all other treatments, 100% mortality was reached with all treatments on the fourth day.

### 3.4. Population and Oviposition Effects of Chips

There was a significant difference between the different treatments in the number of live adults that resulted from continuous population growth for 2 weeks (F = 51.87, df = 4, *p* ≤ 0.0001). There was also a significant effect in the percent larval emergence (F = 21.33, df = 4, *p* ≤ 0.0001), but no significant difference in the number of eggs laid in each treatment (F = 0.328, df = 4, *p* = 0.855). These treatments showed a linear pattern, indicating that with increased treatment there was lower emergence of adult mosquitoes (Figure 4). Female oviposition showed no preference for laying in either treated or control containers. Number of eggs laid in either treated or untreated containers approximated the percent of treated or untreated cups in cage (Table 1), except when similar numbers of treated and untreated containers were placed in the cage with the mosquitoes. In these cages, greater oviposition was observed on the treated containers.

## 4. Discussion

*Aedes aegypti* use different types of containers with varying water volumes [9,15], but the mosquitocidal chips were effective in water varying volumes because the mosquitocidal chips were designed to release 10 PPB pyriproxyfen in 1000 mL of water. In this experiment, varying water volumes would have allowed 10–40 PPB concentrations of pyriproxyfen, if all the active ingredient would have been released from the mosquitocidal chips. Doses as low as 1 PPB of pyriproxyfen result in high mortality of *Ae. aegypti* [11,16,17,18]. Due to pyriproxyfen’s efficacy at such small doses, these chips could be used in larger water volumes to achieve lower concentrations of pyriproxyfen in the water.

*Aedes aegypti* is opportunistic in choosing containers for larval development, therefore, control methods must be adequate for use in different container types from natural to artificial materials. Our results show that the mosquitocidal chips could be used in a variety of containers with minimal differences in *Ae aegypti* mortality. Ceramic and clay containers had the fastest rates of mortality, perhaps due to less absorption of the insecticidal active ingredient to the container walls. This contrasts with studies done by Vythilingam et al. [19], who found that earthen jars reduced long-term efficacy of pyriproxyfen, but those authors found negative effects of these materials after 10 wks. Differences in results may be due to the pyriproxyfen formulation that provides a slow release of pyriproxyfen over longer time. Slow release formulations may be important due to the tendency of some materials to absorb pyriproxyfen [6] reducing its availability in water and mosquito control efficacy.

Pyriproxyfen is also known to tightly adhere onto organic matter [4,20], with consequent decline in concentration in water. However, our experiments demonstrated no significant difference in rate of mortality regardless of the presence of organic matter in the form of oak leaf infusion, which contains mostly leaf chemicals, bacteria, and minimal debris, in contrast suspended organic matter including leaves and soil, which could have more readily absorbed the pyriproxyfen. In contrast with the Schaffer [20] and Sullivan [4] who used ponds containing large amounts of suspended organic debris, the mosquitocidal chips were designed for use in containers around human dwellings where the minimal suspended organic debris would be expected.

## 5. Conclusions

Mosquitocidal chips can serve as an easy-to-use treatment method for *Ae. aegypti* in small containers. When used in a sufficient proportion of artificial or natural breeding containers, these mosquitocidal chips have the potential to reduce mosquito populations in line with results in Sihuincha et al. [11]. The use of pre-treated, slow-release, low-dose pyriproxyfen mosquitocidal chips may prevent potential problems that are associated with high-dose pyriproxyfen use in drinking water [13]. Our studies demonstrated 98% control of *Ae. aegypti* population when all breeding containers were treated. *Ae. aegypti* females use skip oviposition, spreading eggs over multiple water holding containers [8,9], thus it is important that treated breeding sites do not become repellent to mosquitoes, and female *Ae. aegypti* were not deterred to oviposit in cups containing the mosquitocidal chips. The ability of mosquitocidal chips to work for extended periods of time independent of reuse demonstrates their utility to effectively lower populations of *Ae. aegypti.* These mosquitocidal chips have the potential to be an effective, practical and easy-to-use treatment against *Ae. aegypti*.

## Figures and Tables

**Figure 1 ijerph-16-02152-f001:**
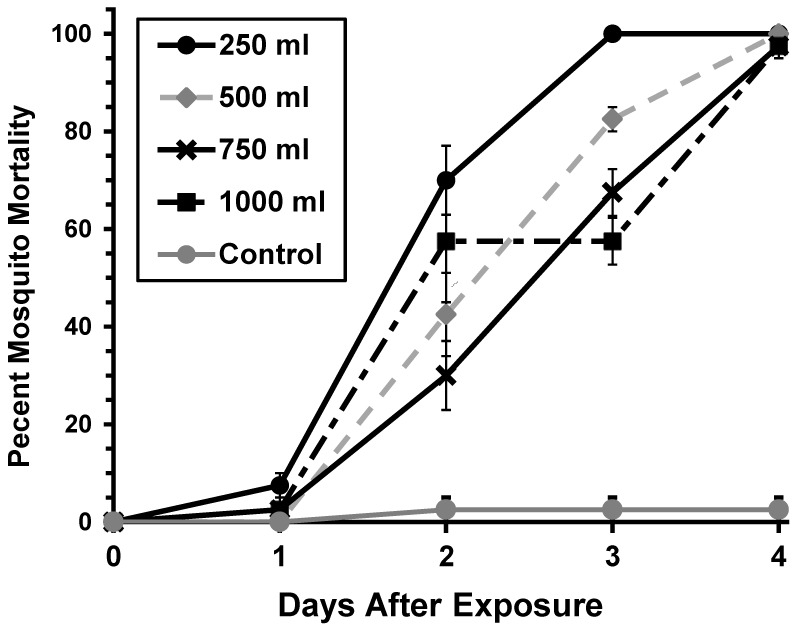
Effects of mosquitocidal chips containing 0.01% Pyriproxyfen on percent mortality of *Ae. aegypti* in varying water volumes. No mortality in untreated controls (not shown). Error bars represent ± standard error of the mean (SEM).

**Figure 2 ijerph-16-02152-f002:**
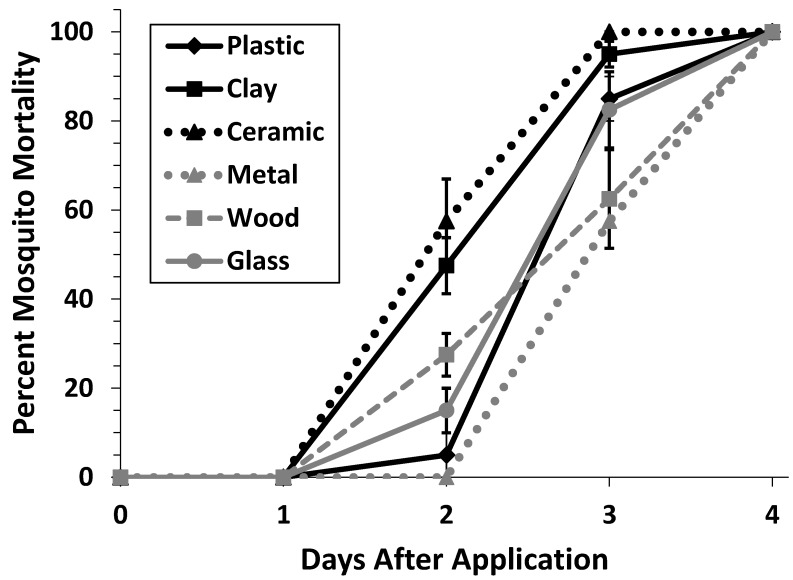
Percent mortality of *Ae. aegypti* when exposed to mosquitocidal chips containing 0.01% Pyriproxyfen in containers of different materials. No mortality was observed in untreated controls (not shown). Error bars represent ± SEM.

**Figure 3 ijerph-16-02152-f003:**
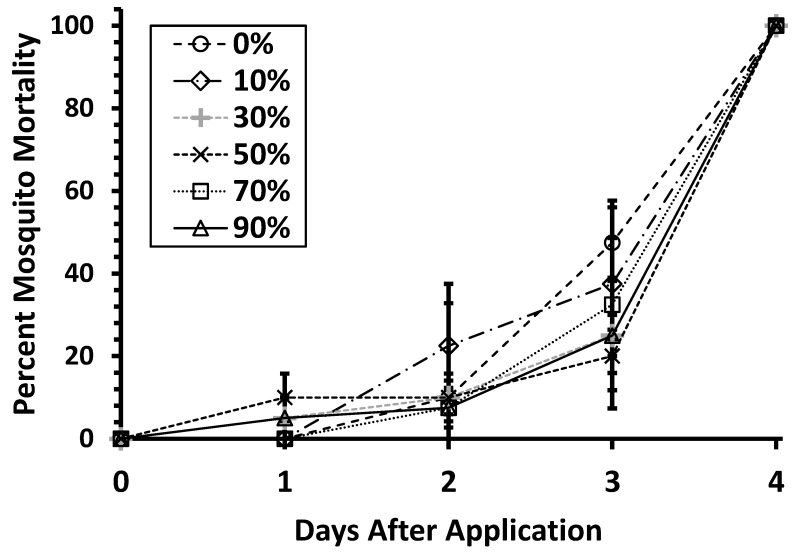
Percent mortality of *Ae. aegypti* when exposed to mosquitocidal chips containing 0.01% Pyriproxyfen in containers with water at different concentrations of organic matter. No mortality was observed in untreated controls (not shown). Error bars represent ± SEM.

**Figure 4 ijerph-16-02152-f004:**
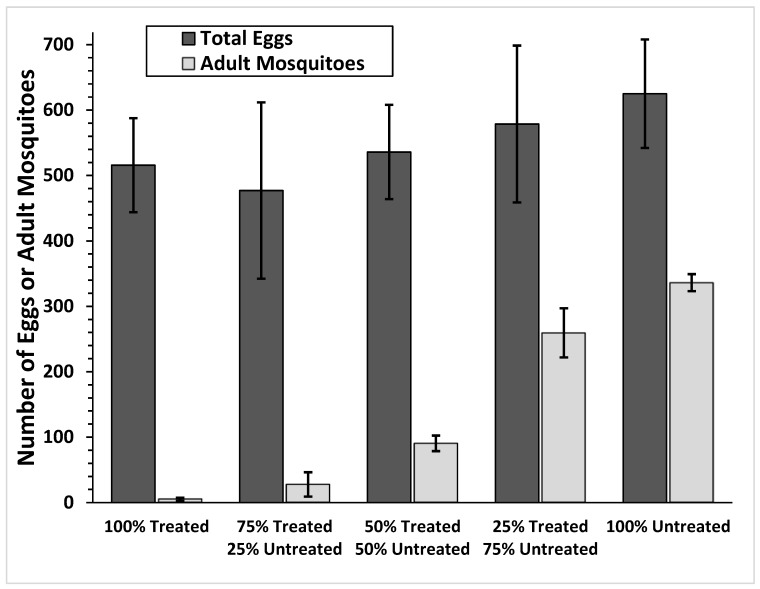
Total number of *Ae. aegypti* eggs laid and adults emerged using five different treatments: (a) 100% Treated—All 4 mosquito breeding cups treated with mosquitocidal chips containing 0.01% pyriproxyfen; (b) 75% Treated 25% Untreated—3 of 4 cups treated with mosquitocidal chips; (c) 50% Treated 50% Untreated—2 of 4 cups treated with mosquitocidal chips, (d) 25% Treated 75% Untreated—1 of 4 cups treated with mosquitocidal chips, and (e) 100% Untreated - none of 4 mosquito breeding cups treated with mosquitocidal chip. Error bars represent ± SEM.

**Table 1 ijerph-16-02152-t001:** Oviposition in pyriproxyfen-treated and untreated water containers in cages with *Aedes aegyptii* females.

Treatment	Treated Containers		Untreated Containers	
	No. of Eggs	% of Eggs	No. of Eggs	% of Eggs
100% TRT	516 ± 71.9	100%	-	-
75% TRT + 25% UT	355 ± 75.8	74%	122 ± 62.5	26%
50% TRT + 50% UT	420 ± 75.7	78%	116 ± 49.6	22%
25% TRT + 75% UT	140 ± 40.5	24%	439 ± 83.7	76%
100% TRT	-	-	625 ± 82.9	100%

TRT = treated; UT = untreated.

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
