# Peer review of "Mosquitocidal Chips Containing the Insect Growth Regulator Pyriproxyfen for Control of Aedes aegypti (Diptera: Culicidae)"

_ijerph, 2019, doi:10.3390/ijerph16122152_

Round 1
Reviewer 1 Report
Authors Stevens, Connelly, Pereira, and Koehler report of the use of mosquitocidal chips with treated juvenile hormone analog, pyriproxyfen, on adult Aedes aegypti mosquitoes. Here researchers assess different effects of (1) water volumes, (2) containers, and (3) organic matter concentrations on efficacy of the mosquitocidal chips. Lastly, the researchers assess the effects of the presence of the mosquitocidal chips on mosquito oviposition, which is highly relevant as the female Aedes aegypti mosquito is known to ‘skip oviposit’.
All in all, this reviewer thinks the data are sound and relevant to the field of mosquito population management. I only have a few minor points as outlined below.
1. In the methods, it is not clear which experiment used 840 µg of pyriproxyfen. As the manuscript reads now, it seems all the experiments used the 8.4 µg pyriproxyfen. Please clarify in methods and discussion sections.
2. Figure legends. While Figure 4’s figure legend is detailed, Figures 1 and 2 are lacking detailed figure legends. Also, Figure 3 does not have a figure legend at all. The authors need to ameliorate this issue by adding more significant descriptions to the figure legends for Figures 1 and 2, as well as including a highly-detailed figure legend for Figure 3.
3. Line 44 in the introduction, change ‘in decreasing’ to ‘to decrease’ for clarity.
4. Line 219 in the discussion, change ‘females uses skip oviposition’ to ‘females use skip oviposition’.
Author Response
Rev1: “In the methods, it is not clear which experiment used 840 µg of pyriproxyfen. As the manuscript reads now, it seems all the experiments used the 8.4 µg pyriproxyfen. Please clarify in methods and discussion sections.”
Reply: Reference to high dose chip was removed because they were not used n experiments reported in this manuscript
Rev1: “Figure legends. While Figure 4’s figure legend is detailed, Figures 1 and 2 are lacking detailed figure legends. Also, Figure 3 does not have a figure legend at all. The authors need to ameliorate this issue by adding more significant descriptions to the figure legends for Figures 1 and 2, as well as including a highly-detailed figure legend for Figure 3.”
Reply: Legends were corrected as suggested.
Rev1: “Line 44 in the introduction, change ‘in decreasing’ to ‘to decrease’ for clarity.”
Reply: Corrected.
Rev1: “Line 219 in the discussion, change ‘females uses skip oviposition’ to ‘females use skip oviposition’”.
Reply: Corrected
Reviewer 2 Report
Stevens et al. demonstrate the mosquitocidal activity of a novel formulation of the juvenile hormone analog, pyriproxyfen. The larvicide was applied to ceramic tiles (called “chips” in this study) in a silica-butyl-methacrylate polymer that allows for its release into water. In order to determine the effectiveness of released larvicide in realistic environments, different vessels for larval growth and amounts of organic material were assessed in larval mortality trials. The effect of different ratios of treated to non-treated breeding cups in a single cage on mosquito survival to adulthood was also considered.
The work clearly demonstrates that Ae. aegypti larvae were killed by pyriproxyfen that was released from the chips. The significance of this research is that it might allow for a convenient delivery system for treating small and isolated peridomestic bodies of water when source reduction is not possible. If it could be demonstrated that consistently low levels of the larvicide (less than 10 PPB) could be released in a controlled manner that is more convenient than existing delivery methods, then this could be a significant innovation.
It was not clear from the text of the introduction or discussion how this system is an advantage over existing delivery systems, such as granules or pellets. Granules containing pyriproxyfen are already being used to treat small bodies of standing water. Is there any inherent advantage to a chip over a granule or pellet? If so, please explain.
The introduction would benefit from an updated assessment of pyriproxyfen in mosquito abatement programs. As mentioned (line 36), the WHO has approved pyriproxyfen at a rate of 10 PPB for use in potable water. However, a recent article co-authored by Dr. Fredrik Nihout (Duke University) provides a cogent argument for a reassessment.
http://currents.plos.org/outbreaks/article/a-possible-link-between-pyriproxyfen-and-microcephaly/
This article is cited by Njoroge and Berenbaum (2019) in the specific context of concerns over its safety in drinking water. As with Dr. Nijhout, Dr. Berenbaum is a highly respected leader in the field of insect science with relevant expertise. A 2016 report from the Swedish Toxicology Research Center identified several areas where sufficient information is lacking in order to perform a suitable risk assessment.” (http://www.diva-portal.org/smash/get/diva2:911791/FULLTEXT01.pdf) Andouze et al. (2018) provide perspective from computer modeling studies on the potential for developmental neurotoxicity via its interaction with vertebrate hormone receptors. Sublethal amphibian toxicity of pyriproxyfen in the range of 10 PPB (10 micrograms/liter) was shown by Lajmanovich et al. (2019). The potential for human developmental neurotoxicity, even if rare and difficult to establish with certainty, cannot be dismissed without a rebuttal of the recent evidence and re-analysis of previous toxicology studies. A mention that concerns have been raised regarding its human safety, and an appropriate reference will suffice.
A potential advantage of the method described in this manuscript is that it could allow for the long-term deployment of low-level pyriproxyfen treatments in peridomestic settings. However, the constant application of pyriproxifen at low levels must be discussed in the context of the evolution of resistance to pyriproxyfen. Resistance was recently reported in natural populations of Aedes aegypti in California (Su et al. 2019) and warrants mention in the introduction. The point can be made without detracting from the study, that pyriproxyfen is one tool in the armamentarium, but it is not a silver bullet.
The expected release from the chips is 10 PPB (Line 189), but the amount of pyriproxyfen is expressed in micrograms per chip (of unspecified dimension). If there is additional work that has shown that this is the expected rate of release, that information should be provided. If a chip coated with 840 micrograms of pyriproxyfen is dissolved in a 350 ml vessel of water, and Line 190 suggests that all of the pyriproxyfen is released from the chip, then the concentration would be 2.4 micrograms per ml, or 2400 PPB. The solubility of pyriproxyfen in water is only 367 PPB (https://www.cdpr.ca.gov/docs/emon/pubs/fatememo/pyrprxfn.pdf), so it would appear that the high dose tiles could potentially release pyriproxifen into the water at a rate that is near its limit of solubility. Results for 840 microgram per chip pyriproxyfen are discussed in the abstact, however, no results are shown for experiments with the 840 microgram chips, so it is unclear why this is even included in the abstract or Materials and Methods section.
By my calculation, the low dose with 8.4 micrograms per chip would be around 24 PPB in 350 ml of water, which is still double the WHO “acceptable” level for human consumption, assuming that it is released from the polymer,
There is no data presented in this manuscript supporting the claim in the abstract that the chips “lasted for extended periods of time”. All of the experiments that I can see that were presented in this study were performed over a period of 4 days.
Likewise, there is no evidence provided to indicate that the 8.4 microgram dose ceased to function after the first week (line 10). More detail would have been necessary for this statement to be interpretable. Does this mean that the chips were not releasing any additional pyriproxyfen when transferred to fresh water after the first week of treatment? This would not be an encouraging result, as doses of ~1 PPB are known to be effective in killing susceptible larvae.
The paper states that the chips were designed to release 10 PPM, but insufficient detail is presented to indicate that this was anywhere near the actual release rate from the chips. The actual dosage of pyriproxyfen in the water was not determined. Although this would not be expected for this preliminary paper, relatively straightforward HPLC methods would allow this to be done https://www.cdpr.ca.gov/docs/emon/pubs/anl_methds/imeth_270.pdf
A more deliberate discussion on the release rate, relating micrograms per chip to parts per billion is necessary and a statement that actual levels of pyriproxyfen were not measured, although they could have been - will suffice.
The experimental setup seems appropriate, though minimal. The WHO guidelines (https://apps.who.int/iris/bitstream/handle/10665/69101/WHO_CDS_WHOPES_GCDPP_2005.13.pdf?sequence=1) for mosquito larvicidal toxicity experiments recommend 25 larvae per trial, in triplicate. This study used 10 larvae per trial. Further “Data from all replicates should be pooled for analysis. LC50 and LC90 values are calculated from a log dosage–probit mortality regression line... …Bioassays should be repeated at least three times, using new solutions or suspensions and different batches of larvae each time.” It is unclear whether different batches of larvae were used in this study. It was unfortunate that the experiment was not designed so that LC50 and LC90 values could be determined.
Ten larvae per trial may be below the recommendations of the WHO. However, this number of larvae was used in recent and comparable studies (e.g., Njoroge and Berenbaum, 2019).
Results of multiple containers could be explained by the transfer of pyriproxyfen from one vessel to another by females. This phenomenon of “autodissemination” has been described in Anopheles arabiensis by Lwetoijera et al. (2019) and should be discussed in the context of the results of this study.
The manuscript is generally well written and grammatically correct and only a few typos were detected
Line 95 missing space Tenlate.
Line 215 “tin”
The figure legends all require additional detail. Dosages, n, and number of replicates for each trial should be included in the figure legends. There is no figure legend for Figure 3.
In summary, this manuscript has potential, but appears to contain vestiges of a larger study that were not trimmed from the abstract. Although mentioned in the abstract and the conclusions, there is no experimental evidence on the high dose treatment or the effectiveness of the chips for "extended periods of time".
Additional details are required to understand the significance of the study, and caveats concerning the use of this insecticide. A single 840 micogram chip in an 8 liter water storage vessel would exceed WHO standards for human exposure.
A drawback of the study is that the actual concentration of the insecticide in the water that contained the larvae was never determined. These concentrations could be quite high - and the lack of any information of how much pyriproxyfen the larvae was exposed to needs to be be addressed in a transparent manner. This is especially important if the authors want to make the case that the chips may be useful in providing a harm-reduced delivery system for pyroproxyfen in certain environmentally appropriate settings.
Author Response
Rev 2: “Stevens et al. demonstrate the mosquitocidal activity of a novel formulation of the juvenile hormone analog, pyriproxyfen. The larvicide was applied to ceramic tiles (called “chips” in this study) in a silica-butyl-methacrylate polymer that allows for its release into water. In order to determine the effectiveness of released larvicide in realistic environments, different vessels for larval growth and amounts of organic material were assessed in larval mortality trials. The effect of different ratios of treated to non-treated breeding cups in a single cage on mosquito survival to adulthood was also considered.
The work clearly demonstrates that Ae. aegypti larvae were killed by pyriproxyfen that was released from the chips. The significance of this research is that it might allow for a convenient delivery system for treating small and isolated peridomestic bodies of water when source reduction is not possible. If it could be demonstrated that consistently low levels of the larvicide (less than 10 PPB) could be released in a controlled manner that is more convenient than existing delivery methods, then this could be a significant innovation.
It was not clear from the text of the introduction or discussion how this system is an advantage over existing delivery systems, such as granules or pellets. Granules containing pyriproxyfen are already being used to treat small bodies of standing water. Is there any inherent advantage to a chip over a granule or pellet? If so, please explain”.
Reply: Added explanations on how the pre-dosed chips facilitates the use of pyriproxyfen in containers
Rev 2: “The introduction would benefit from an updated assessment of pyriproxyfen in mosquito abatement programs. As mentioned (line 36), the WHO has approved pyriproxyfen at a rate of 10 PPB for use in potable water. However, a recent article co-authored by Dr. Fredrik Nihout (Duke University) provides a cogent argument for a reassessment.
http://currents.plos.org/outbreaks/article/a-possible-link-between-pyriproxyfen-and-microcephaly/
This article is cited by Njoroge and Berenbaum (2019) in the specific context of concerns over its safety in drinking water. As with Dr. Nijhout, Dr. Berenbaum is a highly respected leader in the field of insect science with relevant expertise. A 2016 report from the Swedish Toxicology Research Center identified several areas where sufficient information is lacking in order to perform a suitable risk assessment.” (http://www.diva-portal.org/smash/get/diva2:911791/FULLTEXT01.pdf) Andouze et al. (2018) provide perspective from computer modeling studies on the potential for developmental neurotoxicity via its interaction with vertebrate hormone receptors. Sublethal amphibian toxicity of pyriproxyfen in the range of 10 PPB (10 micrograms/liter) was shown by Lajmanovich et al. (2019). The potential for human developmental neurotoxicity, even if rare and difficult to establish with certainty, cannot be dismissed without a rebuttal of the recent evidence and re-analysis of previous toxicology studies. A mention that concerns have been raised regarding its human safety, and an appropriate reference will suffice.
A potential advantage of the method described in this manuscript is that it could allow for the long-term deployment of low-level pyriproxyfen treatments in peridomestic settings. However, the constant application of pyriproxifen at low levels must be discussed in the context of the evolution of resistance to pyriproxyfen. Resistance was recently reported in natural populations of Aedes aegypti in California (Su et al. 2019) and warrants mention in the introduction. The point can be made without detracting from the study, that pyriproxyfen is one tool in the armamentarium, but it is not a silver bullet.”
Reply: Used the reference suggested by the reviewer and discussed how the chips can be beneficial without harming nontarget species and the environment. The present technology could be used with other active ingredients but we are not reporting any results on other a.i.’s here.
Rev 2: “The expected release from the chips is 10 PPB (Line 189), but the amount of pyriproxyfen is expressed in micrograms per chip (of unspecified dimension). If there is additional work that has shown that this is the expected rate of release, that information should be provided. If a chip coated with 840 micrograms of pyriproxyfen is dissolved in a 350 ml vessel of water, and Line 190 suggests that all of the pyriproxyfen is released from the chip, then the concentration would be 2.4 micrograms per ml, or 2400 PPB. The solubility of pyriproxyfen in water is only 367 PPB (https://www.cdpr.ca.gov/docs/emon/pubs/fatememo/pyrprxfn.pdf), so it would appear that the high dose tiles could potentially release pyriproxifen into the water at a rate that is near its limit of solubility. Results for 840 microgram per chip pyriproxyfen are discussed in the abstact, however, no results are shown for experiments with the 840 microgram chips, so it is unclear why this is even included in the abstract or Materials and Methods section.
By my calculation, the low dose with 8.4 micrograms per chip would be around 24 PPB in 350 ml of water, which is still double the WHO “acceptable” level for human consumption, assuming that it is released from the polymer,
There is no data presented in this manuscript supporting the claim in the abstract that the chips “lasted for extended periods of time”. All of the experiments that I can see that were presented in this study were performed over a period of 4 days.
Likewise, there is no evidence provided to indicate that the 8.4 microgram dose ceased to function after the first week (line 10). More detail would have been necessary for this statement to be interpretable. Does this mean that the chips were not releasing any additional pyriproxyfen when transferred to fresh water after the first week of treatment? This would not be an encouraging result, as doses of ~1 PPB are known to be effective in killing susceptible larvae.
The paper states that the chips were designed to release 10 PPM, but insufficient detail is presented to indicate that this was anywhere near the actual release rate from the chips. The actual dosage of pyriproxyfen in the water was not determined. Although this would not be expected for this preliminary paper, relatively straightforward HPLC methods would allow this to be done https://www.cdpr.ca.gov/docs/emon/pubs/anl_methds/imeth_270.pdf
A more deliberate discussion on the release rate, relating micrograms per chip to parts per billion is necessary and a statement that actual levels of pyriproxyfen were not measured, although they could have been - will suffice.”
Reply: Much of the discussion proposed by the reviewer is well beyond the data presented in this manuscript. We have eliminated the mention to the 840 microgram chip and added further discuss on the pyriproxiphen release
Rev 2: “The experimental setup seems appropriate, though minimal. The WHO guidelines (https://apps.who.int/iris/bitstream/handle/10665/69101/WHO_CDS_WHOPES_GCDPP_2005.13.pdf?sequence=1) for mosquito larvicidal toxicity experiments recommend 25 larvae per trial, in triplicate. This study used 10 larvae per trial. Further “Data from all replicates should be pooled for analysis. LC50 and LC90 values are calculated from a log dosage–probit mortality regression line... …Bioassays should be repeated at least three times, using new solutions or suspensions and different batches of larvae each time.” It is unclear whether different batches of larvae were used in this study. It was unfortunate that the experiment was not designed so that LC50 and LC90 values could be determined.”
Reply: Our study was not adequate to produce LC50 and LC90, because it was never designed for that purpose, mianly because we were testing a single done of the insecticide, under different conditions. The objective was to test the effectiveness of this new product under conditions that could be expected in field applications. The methodology to calculate LC50 and LC90 is completely irrelevant for the studies presented in the manuscript.
Rev 2: “Ten larvae per trial may be below the recommendations of the WHO. However, this number of larvae was used in recent and comparable studies (e.g., Njoroge and Berenbaum, 2019).”
Reply: “WHO recommendations are meant to be used in dose response experiments and were never meant to be an absolute requirement for all experiments. For type of experiments we reported in the manuscript, the number of treatments, and replicates we had, 10 insects were an adequate number of insects.”
Rev 2: Results of multiple containers could be explained by the transfer of pyriproxyfen from one vessel to another by females. This phenomenon of “autodissemination” has been described in Anopheles arabiensis by Lwetoijera et al. (2019) and should be discussed in the context of the results of this study.
Reply: We did not want to get into the autodissemination explanation given that the cages we used were relatively small and the close proximity of the different containers
Rev 2: “The manuscript is generally well written and grammatically correct and only a few typos were detected”
Reply: Thank you !
Rev 2: “Line 95 missing space Tenlate.”
Reply: Corrected.
Rev 2: “Line 215 “tin””
Reply: Corrected.
Rev 2: “The figure legends all require additional detail. Dosages, n, and number of replicates for each trial should be included in the figure legends. There is no figure legend for Figure 3.”
Reply: Corrections were made to address the comments.
Rev 2: “In summary, this manuscript has potential, but appears to contain vestiges of a larger study that were not trimmed from the abstract. Although mentioned in the abstract and the conclusions, there is no experimental evidence on the high dose treatment or the effectiveness of the chips for "extended periods of time".
Additional details are required to understand the significance of the study, and caveats concerning the use of this insecticide. A single 840 micogram chip in an 8 liter water storage vessel would exceed WHO standards for human exposure.
A drawback of the study is that the actual concentration of the insecticide in the water that contained the larvae was never determined. These concentrations could be quite high - and the lack of any information of how much pyriproxyfen the larvae was exposed to needs to be be addressed in a transparent manner. This is especially important if the authors want to make the case that the chips may be useful in providing a harm-reduced delivery system for pyroproxyfen in certain environmentally appropriate settings.”
Reply: The reviewer is correct. References to the larger study (MS thesis by first author) were eliminated. We also removed reference to the high dose chip and restricted discussion of potential pyriproxyfen concentration in water in relation to the low (8.4 microgram) dose. Discussion on potential issues associates with the use of the mosquitocidal chips were included in the manuscript but were kept to a minimum since this is not a field study. Further discussion will be included in the discussion of field work at the appropriate time
Thanks
Roberto Pereira
Round 2
Reviewer 2 Report
The revised manuscript is a major improvement. Almost all of my concerns were eliminated with by removing mention of the high dose treatment. Now it is entirely clear they are not looking at a dose-response issue, so I agree that LC50, LC90 is no longer relevant. This was not as clear in the previous version.
The addition of recent references on the need to reassess the safety of pyriproxifen is appreciated.
I agree that the other issues (autodissemination and resistance) are beyond the scope of the data. The role of this delivery system in the evolution of resistance is relevant and I hope it will be addressed in forthcoming field studies.
The authors have done an excellent job in addressing all of my concerns and I recommend publication.